# Validation of the Polish version of the Brief Resilience Scale (BRS)

**Karol Konaszewski**[1]*, **Małgorzata Niesiobędzka**[2], **Janusz Surzykiewicz**[3]

**1** Faculty of Education, University of Bialystok, Bialystok, Poland, **2** Faculty of Philosophy and Education, Katholische Universität Eichstätt-Ingolstadt, Eichstätt, Germany, **3** Faculty of Education, Cardinal Stefan Wyszynski University in Warsaw, Warsaw, Poland

* k.konaszewski@uwb.edu.pl, karolkonaszewski@wp.pl

**Data Availability Statement:** All data files are available from the OPEN ICPSR database: https://doi.org/10.3886/E119441V1.

**Funding:** The author(s) received no specific funding for this work.

## Abstract

### Background

We conducted three studies to validate the Polish version of the BRS. Our objectives are as follows: first, to explore the dimensional structure of the scale and to determine the internal consistency (study 1: n = 1022); second, to determine the congruent and divergent validity of the BRS (study 2: $n$ = 242); and third, to examine sensitivity of the BRS scale to detect high-risk population (study 3: $n$ = 602).

### Methods

To explore the dimensional structure of the scale, we tested a two-factor model with one factor for positively worded items and one factor for negatively worded items. To determine the congruent and divergent validity of the BRS, we analysed correlations among BRS and resilience, positive mental health, and with positive and negative religious coping. We used Student's t-test to examine sensitivity of the BRS scale to detect a high-risk population.

### Results

Based on the CFA, a bivariate model was confirmed for items positively and negatively formulated with a higher order factor, which indicates the homogeneity of the scale, similar to the analyses carried out for their language versions confirming this type of homogeneity of the scale. The internal compatibility assessment based on Cronbach's Alpha and McDonald's Omega is good (0.88). Our analyses intended to test convergent and divergent validity, and showed that the BRS results are significantly related to a questionnaire measuring similar constructions. Our validation studies also provided important diagnoses regarding BRS "sensitivity", indicating that groups with higher stress levels achieved lower BRS resilience results.

### Conclusion

The results of our research indicate that the Polish version of the BRS should be considered to be a reliable and valid research tool. The Polish version of BRS is a reliable and accurate way of measuring resilience as the ability to bounce back from adversity and overcome

**Competing interests:** The authors have declared that no competing interests exist.

various challenges or stressors. This scale may be used for both research and intervention purposes.

## Introduction

The term resilience—as a characteristic phenomenon describing the property or the ability to adapt or adapt well despite challenges and difficulties or losses—is becoming increasingly more interesting for researchers and practitioners of the psychosocial functioning of people and their health. Research projects require diagnostic tools that are reliable and economic. The diversity of these tools is associated with the sole understanding and defining of resilience, including complex and multidimensional theoretical and conceptual approaches that are related to the needs of a specific field of knowledge. This results in various operationalizations of the given constructs and their measurement. Depending on the adopted conceptual approaches, manner of describing, explaining, and determining the definition boundaries, the following characteristic trends of resilience can be distinguished: *resilience as a trait* [1–5], as *an ability* [6–10], as *a process* [11–17], and as *an outcome* [18–21].

Resilience, when treated as a trait, is related to health and effectively coping with the problems of personal and social life. It is a positive, individual feature of a given person that works in favor of the process of adaptation, prevents the negative effects of experienced stress, and minimizes depression. Resilience is mainly associated with the possibility of diagnosis and registration of factors that affect the development of a characteristic predisposition of an individual for immunity, flexibility, and so on. In other words, it allows for the prospective determination of features and their relation with health, or adaptation and coping with stress [2,4,22,23]. This group of researchers also emphasize certain skills that lead to effective coping [24–26]. In this context, resilience is more than just a property (trait) of the personality structure. Rather, it is a human-specific ability to organize and develop one's own activity in changing living conditions by proper "grounding" in the environment, which includes a relatively clear "horizon" of references and orientation. This concept refers to the human individual as a living whole; that is, as a specific person perceiving their own capabilities and limitations in a given life situation [27]. In this approach, resilience constitutes a set of abilities for dealing with difficulties and problems, the ability to flexibly approach difficulties, and the ability to create and maintain satisfying social relations [28]. Following the development of research on the phenomenon of resilience, attention began to focus not only on the key properties of individuals or factors associated with it but also on understanding the mechanisms by which they operate. In this approach, resilience has been defined as a dynamic process that reflects a relatively good adaptation—a positive adaptation of an individual, despite the experienced threats, adversities, or traumatic experiences. This process involves the interaction of an entire spectrum of risk factors, vulnerabilities, and protective factors [14–17,29]. The approach to defining and understanding resilience also fits in the category of an outcome. Understanding resilience can be read in the category of outcome, and as a result conditioned by many factors. This approach to resilience places emphasis on the importance of individual and environmental features that interact. In this case, resilience is understood as the outcome of this interaction. The interaction between an individual and various contextual aspects leads to attitudes that produce lasting positive results with a continuous learning process to renew and balance the situation [21,30–32].

Researching abilities in the field of resilience more closely is important from both a research and theoretical point of view, and also because of its practical terms. This important conceptual aspect of resilience allows us to determine the abilities of an individual to effectively adapt to life tasks in the face of an unfavorable life situation or highly adverse conditions. There is also a need for the possibility of diagnosing or measuring this type of an individual's abilities when confronted with various challenges, or its connection with other immune resources and positive health indicators, such as life satisfaction, wellbeing, or negative health indicators (e.g. assessment of depression or perceived stress). A closer definition of the ability of an individual to "bounce back" from negative experiences related to family or relational problems, health problems, or problems in the workplace increases when we deal with different social contexts and the specifics of family life [26,33–39].

## Demand for diagnostics

The possibility of an adequate and relatively economic measurement of resilience as a "ability to bounce back" has been shown in recent years to a particularly large degree by the BRS scale [9]. Consequently, the main purpose of this study is to assess the psychometric properties of BRS in a Polish population of subjects. There are many methods for measuring resilience. Some researchers use tools focused on measuring "immune resources", such as the General Self-Efficacy Scale [40] or the Sense of Coherence Scale [41,42]. Whereas, others use tools have been specifically designed to measure resilience. Among others, Ahern and colleagues, and Windle and colleagues reviewed resilience scales within different age groups by assessing their psychometric properties [32,43]. These authors found that among the psychometrically reliable and conceptually justified scales, special attention should be placed on the Brief Resilience Scale (BRS), which—together with, among others, the Resilience Scale (RS) and Connor—Davidson Resilience Scale (CD-RISC)—is one of the internationally most widely-used scales, has been validated in many languages, and has been shown to have outstanding psychometric properties. Furthermore, resilience relating to the traits and properties of individuals has primarily been measured with scales such as the Connor-Davidson Resilience Scale (CD-RISC) [44], the Dispositional Resilience Scale (DRS) [45,46], the Ego-Resiliency Scale (ERS) [1], Psychological Resilience (PR) [29], the Resilience Scale for Adults (RSA) [47] and the RS [22]. However, very few tools have focused on aspects of the Resilience Process Questionnaire (RPQ) [48]. In addition, efforts have been made to determine measures of resilience understood in terms of outcomes (e.g. educational) [19]. These scales are characterized by good psychometric properties, validity, and reliability. While some of them try to determine the severity of resilience understood as a trait or process, others focus on factors that comprehensively determine resilience taking into account personality and social categories. All of these tools have been documented in numerous studies, which analysed resilience as a predictor of many negative and positive results [23,49–51] and attempts have been made to state what may determine the resilience of individuals, groups, and communities [52,53]. As already indicated in case of these scales, resilience is defined as a property-trait of units that work in favor of adaptation, such as after stress [4] or as a process of dealing with trauma, overcoming its use of negative aspects as elements of individual growth [54]. All these tools primarily assess the traits, properties, and resources of individuals that determine resilience, only in some cases have attempts been made to refer to processual aspects or resilience results.

Polish research has mostly taken advantage of the Ego-Resiliency Scale, which was adapted by Kaczmarek (2011), and the RS adapted by Surzykiewicz and colleagues (2019). The Psychological Resilience Scale (SPP-25) by Ogińska-Bulik and Juczyński (2011) is often used, as is its shorter version, the SPP-18, which is intended for measuring resilience in children and

adolescents [55]. Other tools are also used, such as the KOP-26 Resilience Assessment Questionnaire by Cechowski and colleagues [56] and the Ego Questionnaire of Feeling Safe and Resilient (KPB-PE) [57], which (similar to other scales) focus on resilience mainly understood as a trait or have included the conglomerates of protective factors, both individual and environmental.

Developing a measuring possibility that views resilience as the ability of individuals to resist or "bounce back from adversity" is very important and desirable in terms of research, both for the development of scientific knowledge and for its practical application. Understanding resilience as the ability to bounce back or recover from stress can be important for assessing the positive and negative indicators of the functioning of individuals and, in particular, for designing activities aimed at shaping resilience when understood in terms of an ability. Furthermore, this ability may be extremely important for people who are already sick or have to deal with constant stress related to health. Therefore, the validation of a proven and valuable tool in this field—that is, BRS [9]—under Polish conditions will help to develop an important conceptual and diagnostic research area.

## Psychometric properties of BRS

To accurately assess resilience, meaning an individual's ability to recover from stress despite significant adversity (e.g. chronic stressors or adverse life events), Smith and his colleagues developed the BRS. The theoretical basis of their scale includes the primary English understanding of "resilience," in which the word "resile" means "bounce or spring" (from re- "back" + salire- "jump, jump") [9,58]. In the case of the BRS, "resilience" has been defined in a variety of ways, including the ability to bounce back or recover from stress, to adapt to stressful circumstances, to not become ill despite significant adversity, and to function above the norm in spite of stress or adversity [9].

The BRS consists of six items and was created to assess the ability to bounce back or recover from stress. Its psychometric characteristics were examined in four samples: two student samples, and patients with heartache and chronic diseases. The respondents used a 5-point Likert scale to rate to what extent they agree with a given statement (1 strongly disagree, 5—strongly agree). The authors of the scale showed that BRS negatively connects with anxiety, depression, negative affect, and perceived stress [9]. To date, several studies have adapted BRS. A univariate solution was confirmed in Spanish studies. The psychometric properties of its scores were examined in a heterogeneous sample of 620 Spanish adults. Validity was confirmed by analysing the relation between BRS and coping, post-traumatic growth, anxiety, depression, or perceived stress [59]. In the Malaysian version, based on an exploratory factor analysis, a univariate solution with reliability was also demonstrated ($\alpha$ = .93) [60]. Validation studies have also been carried out in the German population to assess the structure, validity, and reliability of data that was collected from the population in two vast samples, including one representative. The method-factor model showed an excellent model fit, which was significantly better than the one-factor model or the two-factor model. The validity was measured by correlating BRS results with measures of mental health, wellbeing, coping, social support, and optimism [61]. A univariate solution has also been confirmed in studies of students from Hong Kong and mainland China. The BRS was found to measure one single construct and exhibited convergent validity in both samples [7]. As part of two studies, a Brazilian adaptation of BRS was performed. Confirmatory factor analysis (CFA) supported the predicted one-factor solution, with five items and satisfactory internal consistence (a = 0.76) [62]. A Romanian study has also adapted BRS, where the scale was applied in two different samples at different times. In the first study, which was conducted on a sample composed of 198 military students, factor

analysis revealed the presence of a single factor, that was weakly charged with five of the six items of the scale. In the second study, which was conducted on one sample made up of 166 employees in the Romanian military system, all of the items were satisfactorily loaded on a single factor and the Cronbach's alpha value indicated good internal consistency of the scale [63].

Studies have confirmed the high accuracy of the BRS scale [9,10,59,61]. However, it is worth emphasizing here that research concerning the predictive validity of the scale is scarce [59]. The predictive validity of BRS in detecting groups experiencing severe stress was one of the elements of working on the Spanish version of BRS [59]. Studies carried out among a group of parents of sick children, sick people, and the general population did not fully confirm the sensitivity of BRS in detecting groups experiencing severe stress. This study showed only significant differences in the severity of resilience between a group of parents of critically ill children and a group of cancer patients. The hypothesis assuming the lowest level of resilience in the heavily stressed group was not confirmed [59]. In our opinion, it is worth further exploring this direction of research to confirm the predictive accuracy of BRS in detecting groups experiencing severe stress.

Based on these findings, it should be stated that BRS possesses good psychometric properties that can confirm a univariate structure regardless of cultural aspects. The reliability of the scale is at least good, and its accuracy has been confirmed by analysing the correlations between BRS and other measures of resilience, as well as between BRS and health indicators related to stress or personality traits [9]. Due to the good psychometric properties and the economic structure, it was also decided to adapt the tool to Polish conditions.

**Polish validation of BRS.** We conducted three studies to validate the Polish version of the BRS. Our objectives are as follows: first, to explore the dimensional structure of the scale and to determine the internal consistency (study 1: n = 1022); second, to determine the congruent and divergent validity of the BRS (study 2: $n$ = 242); and third, to examine sensitivity of the BRS scale to detect high-risk population (study 3: $n$ = 602).

To explore the dimensional structure of the scale, we tested a two-factor model with one factor for positively worded items and one factor for negatively worded items. To determine the congruent and divergent validity of the BRS, we analysed correlations among BRS and resilience (RS-14; Wagnild & Young, 1993), positive mental health, and with positive and negative religious coping [64]. We used Student's t-test to examine sensitivity of the BRS scale to detect a high-risk population. In line with Smith and colleagues results, we expected that groups under higher levels of stress would score lower on resilience, and thus we predicted higher level of resilience in the group of parents of children without special educational needs than in the group parents of children with special educational needs [9].

## Study 1: Factor structure and reliability of the scale

### Materials and methods

**Participants.** The first study was conducted on a Polish national representative sample. The sample consisted of 1022 adults who were aged 18–88 years ($M$ = 44.55, $SD$ = 15.54). Around half of the participants were women (52.4%). Almost one third had a postgraduate degree (31.7%), 9% had a bachelor's degree, 45% of the sample finished education on secondary level, and 14.3% finished on primary level. Regarding age, 12.6% of the sample was in the age interval between 18–24 years, 20.5% between 25–34 years, 16.3% between 35–44 years, 18.5% between 45–54 years, and 32.1% were above 55 years old.

**Procedure and ethics statement.** The study was conducted on a Polish national representative sample, recruited on an on-line panel by research agency. Computer-assisted web interviewing (CAWI) technique was used in which the interviewer follows a script provided in a

website. After providing informed consent, the system confirmed data confidentiality, and informed the respondents that participation was voluntary and that they could withdraw at any time. The data were collected on December 2019. The Ethics Committee of the Education Faculty at the University of Bialystok approved the study, which was carried out in accordance with the Board's recommendations.

**Materials.** *The Brief Resilience Scale*. The BRS is formed by six items with a 5-point Likert response scale, ranging from 1 = strongly disagree to 5 = strongly agree. Three items are positively phrased and the other three are negatively phrased. The BRS is scored by reverse coding items 2, 4, and 6, and then calculating the mean of the six items. The original English version of the BRS demonstrated good internal consistency with a Cronbach's alpha value, ranging from .80 to .91 [9].

The original version the RBS was translated into Polish by four independent translators with a high proficiency in English. The translations were adjusted to the final version of the scale by three of the authors of the present study. Next, the final version was back-translated into English by two independent translators with a high proficiency in English. Any differences between the original and back-translated version of the BRS were discussed and amended by three authors of the study and the final version of the RBS was accepted by the author of the scale. The translation of the scale was done according to accepted principles developed for the purposes of intercultural research (WHOQOL Translation Methodology), based on the original English version.

**Data analysis.** Item endorsements in each response category and corresponding skewness and kurtosis values were calculated. Next, the factor structure and reliability of the BRS was examined. CFA with maximum likelihood (ML) estimation implemented in AMOS 24 was applied to assess the factor structure of the scale. The ML method was used due to small deviations from a normal distribution and the representative size of the study group. The chi-squared statistic was used to assess the sample and the implied covariance matrices; however, this statistic strongly depends on sample size and provides an overly conservative assessment of model fit. The comparative fit index (CFI) and the goodness-of-fit index (GFI) were used to assess model fit relative to a baseline model in which all variables are uncorrelated and values above .95 indicate good fit, while values above .90 are considered to indicate acceptable fit. The root-mean-square error of approximation (RMSEA) was also presented. Ideally, this should be less than .05, but values less than .08 are considered to be acceptable [65–67]. We tested a two-factor model with one factor for positively worded items and one factor for negatively worded items. We decided to test this model due to the results of previous studies on the adaptation of the BRS scale. The studies indicate that the parameters of the model with two higher factors has the best fit [59,61]. Furthermore, we considered wording effect emphasized by Rodríguez-Rey, Alonso-Tapia & Hernansaiz-Garrido [59]. The effect demonstrates that the positively and negatively worded items often establish two factors even when the content of these items is consistent. For the reliability, we calculated Cronbach's α and also the composite reliability (McDonald's omega ω).

## Results

**Distribution of scores.** The data demonstrated that item distribution covered the full spectrum of response categories. In addition, skewness and kurtosis values were within the range to consider the BRS items' responses as not deviating from normality. Table 1 presents the distribution of responses in each BRS item, and the levels of skewness and kurtosis. The mean total score of the BRS was 18.18 ($SD$ = 4.50), with an observed range between 6 and 30, a skewness value of $S$ = .002, and a kurtosis of $K$ = .06.

**Table 1. Item endorsements in each response category and corresponding skewness and kurtosis values.**

|  |  | Response | Category | N (%) |  |  |  |
|---|---|---|---|---|---|---|---|
| Item | 1 | 2 | 3 | 4 | 5 | Skewness | Kurtosis |
| BRS1 | 36(3.5) | 154(15.1) | 393(38.5) | 379(37.1) | 60(5.9) | -.36 | -.13 |
| BRS2(R) | 54(5.3) | 311(30.4) | 359(35.1) | 251(24.6) | 47(4.6) | .10 | -.59 |
| BRS3 | 47(4.6) | 260(25.4) | 365(35.7) | 305(29.8) | 45(4.4) | -.09 | -.59 |
| BRS4(R) | 49(4.8) | 312(30.5) | 387(37.9) | 226(22.1) | 48(4.7) | .17 | -.45 |
| BRS5 | 48(4.7) | 238(23.3) | 411(40.2) | 290(28.4) | 35(3.4) | .-15 | -.42 |
| BRS6(R) | 44(4.3) | 276(27) | 384(37.6) | 268(26.2) | 50(4.9) | .03 | -.52 |

*1 = Strongly Disagree 2 = Disagree 3 = Neutral 4 = Agree 5 = Strongly Agree*

**Factor structure.** The results confirm that the model with two first-order factors for positively and negatively worded items provided an excellent fit to the data: $\chi2(8) = 14.90$; $p < .06$; $\chi2/df = 1.86$; RMSEA = .029 (low = .001; high = .052); GFI = .995; CFI = .998. Factor loadings were high and exceeded a magnitude of .76. Fig 1 demonstrates the standardized estimates of the confirmatory model.

**Reliability of the BRS.** Cronbach's alpha coefficient demonstrated good reliability of the BRS, with $\alpha = .88$. The composite reliability was also good, with McDonald's omega $\omega = .88$, which indicates the proportion of a scale's variance due to a unidimensional factor.

**BRS and demographic variables.** The t-test for independent samples confirmed that there was relationship between the results obtained on the resilience and gender ($t(1020) = -5.61$; $p < 0.05$). Analysis indicated that men's ($M = 18.99$; $SD = 4.22$) resilience scores were higher than the women's ($M = 17.43$; $SD = 4.61$). The r-Pearson correlation showed a low, positive correlation between the results on the RS and age ($r = .17$; $p < .001$).

## Study 2: Congruent and divergent validity of the scale

### Materials and methods

**Participants.** The sample consisted of 242 students, who were aged 20–29 years ($M = 23.43$, $SD = 4.99$). Most of the participants were women (64.05%).

**Procedure and ethics statement.** The participants were recruited via the Internet. An e-mail was sent to the students to explain the purpose of the research, data confidentiality, and it informed respondents that participation was voluntary and that they could withdraw at any time. The message also contained a link to the online questionnaire and access passwords to complete the questionnaire, after providing informed consent. The data were collected on March 2020. The Ethics Committee of the Education Faculty at the University of Bialystok approved the study, which was carried out in accordance with the Board's recommendations.

**Materials.** *The BRS*. The Polish version of the BRS tested in the study 1 was used to assess resilience.

*Resilience Scale RS-14*. Resilience was also measured by the RS-14) [22] in Polish adaptation made by Surzykiewicz. Konaszewski and Wagnild (2019). The authors of the scale defined resilience as a personality characteristic that supports the process of adaptation in difficult situations. The scale consists of 14 items. The respondents were asked to rate the degree to which they agree or disagree with each item on a 7-point Likert scale, from 1 ("I do not agree") to 7 ("I agree"). The Polish version of the RS-14 had shown test-retest to have a very good reliability (.88) and a good internal consistency ($\alpha = .85$) [68].

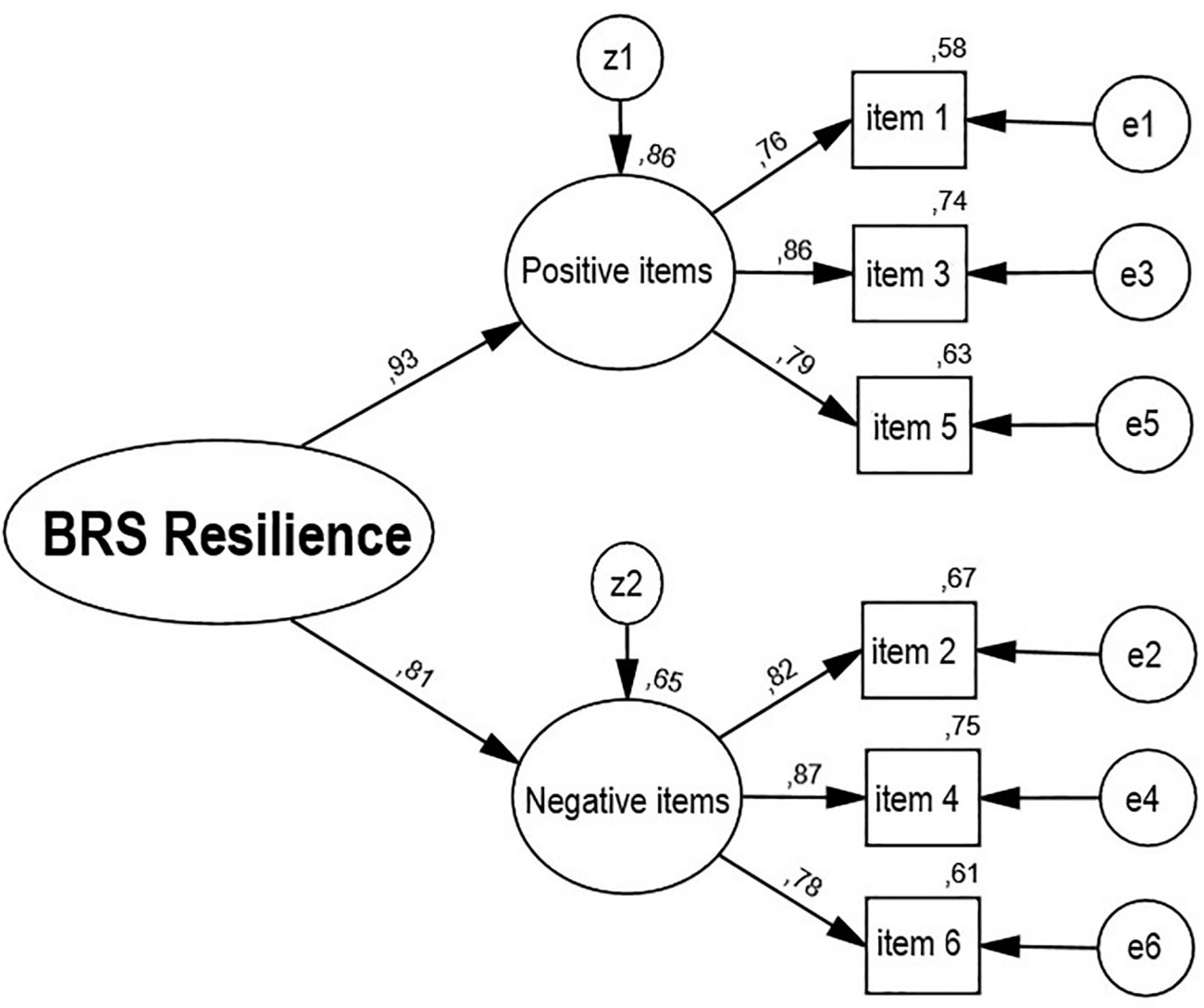

**Fig 1. Factor analysis of the BRS.**

*Warwick-Edinburgh Mental Wellbeing Scale.* Mental wellbeing was measured with the Warwick-Edinburgh Mental Wellbeing Scale (WEMWBS; [69]) in Polish adaptation made by Konaszewski, Niesiobędzka and Surzykiewicz (under review). The scale consists of 14 positively worded items relating to positive mental wellbeing. The participants estimated their feelings and thoughts over the last 2 weeks on a 5-point Likert scale, ranging from 1 ("none of the time") to 5 ("all of the time"). The Polish version of the WEMWB had a very good internal consistency ($\alpha = .92$). The Polish version of the WEMWBS had shown test-retest to have a very good reliability (0.87).

*Religious Coping.* To measure religious coping, we used RCOPE [64]. The religious strategies under consideration represent a broad spectrum of the studied reality and they include positive and negative, passive, active, and interactive strategies relating to God and the Church. The complexity and multidimensionality of the phenomenon has been confirmed by the results of a factor analysis. The Polish version of the RCOPE consists of 105 items on 16 scales, with both positive [9] and negative [7] religious strategies, but only 85 items are diagnostic.

**Table 2. Pearson's correlations BRS with resilience (RS-14), wellbeing, religious positive coping and religious negative coping.**

| Variable | BRS | RS 14 | WB | RPC |
|---|---|---|---|---|
| Resilience (RS 14) | .500*** | - | | |
| Wellbeing (WB) | .466*** | .732*** | - | |
| Religious Positive Coping (RPC) | -.083 | .044 | .097 | - |
| Religious Negative Coping (RNC) | -.184** | -.170** | -.004 | .311*** |

* p < .05,

** p < .01,

*** p < .001

Participants respond to the items on a 4-point scale, with their answers ranging from 0 (never) to 3 (always). They assess the degree to which they make use of various religious coping strategies, both positive (Cronbach's α 0.91) and negative (Cronbach's α 0.71) [70].

**Data analysis.** To determine the congruent of the BRS, we analysed correlations among BRS and resilience measured with RS-14) [22], wellbeing [71], and with positive and negative religious coping [64]. In line with the previous results, we expected positive relationships BRS with different resilience measures [9,59], with wellbeing [51,72], and with positive religious coping [73,74]. Furthermore, we expected a negative relationship between BRS and religious negative coping [73,74].

## Results

As expected, BRS strongly and positively correlated with resilience measured by RS-14 ($r = .50$, $p < .001$) and moderately with wellbeing ($r = .46$, $p < .001$). Moreover, the relationship between BRS and religious negative coping was significant and negative ($r = -18$, $p < .01$). No significant correlation was noticed between BRS and religious positive coping ($r = -08$, $p > .05$) (Table 2).

## Study 3: Sensitivity of the scale

### Materials and methods

**Participants.** The sample consisted of 602 adults aged 20–86 years ($M = 38.17$, $SD = 10.64$). The sample included parents of children without special educational needs ($n = 310$, 65.4% women, $M = 35.19$, $SD = 10.03$) and parents of children with special educational needs ($n = 292$, 64.7% women, $M = 41.30$, $SD = 10.36$). The latter consists of parents of intellectually disabled children ($n = 107$), parents of blind and visually impaired children ($n = 5$), parents of deaf and hearing impaired children ($n = 6$), parents of autistic, including Asperger syndrome, children ($n = 59$), parents of physically disabled children ($n = 21$), parents of children with multiple impairments ($n = 2$), parents of children with ADHD ($n = 2$), parents of socially maladjusted children and with risk of social maladjustment ($n = 23$), parents of children with communication disability ($n = 64$), and parents of children with specific learning difficulties ($n = 3$).

**Procedure and ethics statement.** This study was conducted through direct contact with the respondents. The respondents received a research sheets in paper form and were informed about the purpose of the study, their voluntary participation, and their ability withdraw at any time. The data were collected between December 2019 and March 2020. The Ethics Committee of the Education Faculty at the University of Bialystok approved the study, which was carried out in accordance with the Board's recommendations.

**Materials.**   *The BRS.* The Polish version of BRS tested in Study 1 was used to assess resilience.
*Perceived stress.* was measured with Perceived Stress Scale [75] in Polish adaptation made
by Juczyński and Ogińska-Bulik (2012). The scale is made up of 10 statements on a 5-point
response scale, ranging from 0 (never) to 4 very often. The Polish version of the PPS has
shown a good internal consistency: Cronbach's alpha was .86 [76]. This scale also showed
good internal consistency in our sample ($\alpha$ = .81).

**Data analysis.**   First, we used Student's *t*-test to examine differences in perceived stress
between group of parents of children with and without special educational needs. Next, we
examined the sensitivity of the BRS scale to detect a high-risk population using Student's *t*-test.
In line with Smith and colleagues results, we expected that groups under higher levels of stress
would score lower on resilience, and thus we predicted higher level of resilience in the group
of parents of children without special educational needs than in the group parents of children
with special educational needs [9]. Effect sizes were evaluated with Cohen's *d*: effects with *d* =
.2 to .5 were interpreted as small, effects with *d* = .5 to .8 were considered medium, and effects
with *d* > .8 were considered large.

## Results

Significant differences were found in perceived stress between groups (*t*(600) = -3.28; *p* <
.001). Parents of children with special educational needs experienced stress to a larger extent
(*M* = 18.89, *SD* = 6.23) than parents of children without special educational needs (*M* = 17.13,
*SD* = 6.88). We expected that the group under a higher level of stress would score lower on
BRS. The results of Student's *t*-test demonstrated significant differences in the level of resil-
ience between groups (*t*(600) = 5.71; *p* < .001; Cohen's *d* = .47, a small effect). The parents of
children with special educational needs showed a lower degree of resilience (*M* = 17.94,
*SD* = 5.99) than parents of children without special educational needs (*M* = 20.51, *SD* = 5.00).

## Discussion

The results of our research indicate that the Polish version of the BRS should be considered to
be a reliable and valid research tool. The BRS, constructed by Smith and his research colleagues
(2008), is a popular tool for measuring resilience [9]. This tool results from a demand for
methods defining the described construct, expressing an individual's abilities dispositional
properties to predict behavior, and also psychosocial and health functioning. Based on the con-
ducted research, it can be concluded that the Polish version of BRS does not differ in its value
from the original version and it possesses good psychometric properties. In Polish conditions,
the BRS is an accurate way to assess resilience as the ability to resist in the sense of "resiliently
bouncing back" or adapting to challenges, or recovering after stress. It can also provide impor-
tant information on an individual's coping and functioning among various stressors, especially
those related to health. Our results also confirm the documented psychometric properties [9],
as well as similar results from numerous international validation studies [7,59,61,63] and
meta-analyses [16,77].

Based on the CFA, a bivariate model was confirmed for items positively and negatively for-
mulated with a higher order factor, which indicates the homogeneity of the scale, similar to the
analyses carried out for their language versions [33,59,63], confirming this type of homogene-
ity of the scale. This type of model with two factors, first order, for positively and negatively
formulated positions is very well fitted to the data. Our research results confirm that the inter-
nal compatibility assessment based on Cronbach's Alpha and McDonald's Omega is good
(0.88) and close to those obtained by the authors of the scale and in the presented adaptive
studies [9,59,61]: the obtained values prove the good internal reliability of the questionnaire.

Moreover, our analyses intended to test convergent and divergent validity, and showed that the BRS results are significantly related to a questionnaire measuring similar constructions. The assessment of the theoretical value of the BRS resilience construct in relation to the related concept of RS 14 shows the external consistency of the construct, replicating its theoretical validity. As expected, a positive correlation was obtained in the resilience index as measured by BRS and resilience as an individual-personality trait of an individual measured with the RS-14. Furthermore, our studies have also confirmed the positive relationship between resilience and wellbeing, as well as a negative relation between resilience and negative religious coping. However, the relationship between resilience and positive religious coping has not been confirmed, as in studies by Jans-Beken (2019) [78] and Mohr with colleagues [79]. It is worth noting that researchers frequently do not achieve clear links between resilience and religious coping [80–82]. Furthermore, unlike previous research [59], our results confirmed the sensitivity of the scale in terms of detecting groups under high stress. Parents of children with special educational needs exhibited a significantly lower resilience than parents of children without special educational needs.

Our analysis of the research results confirmed the existence of differences in terms of severity of resilience among the group of women and men. The analysis showed that the intensity of resilience was higher among men than within the group of women, similarly to studies by Boardman, Blalock, and Button [83]. Our study also showed a low, positive correlation between BRS results and age. The obtained results are in line with previous findings, indicating quite clearly that resilience tends to increase during the life cycle [84], acting (for example) as a mediator in order to achieve positive adaptation at subsequent stages of life.

Our validation studies also provided important diagnoses regarding BRS "sensitivity", indicating that groups with higher stress levels achieved lower BRS resilience results. Thus we have predicted a higher resilience level in the group of parents of children with no special educational needs than in the group of parents with children who have special educational needs. Considering that BRS constitutes a characteristic measure of resilience—which specifically assesses it in its original and most basic meaning: bouncing back or recovering after stress [9], and not only as personality traits or protective factors expressing a specific predisposition [32]—it can be assumed that BRS brings an important cognitive and practical aspect for both diagnosis and intervention-activities. Our research results, to some extent, also allow us to confront the question concerning the value of measuring constructs in relation to rehabilitation and therapeutic activities, where the question of whether we are dealing with resilience as a personality trait or skill is of great importance due to the possibility of adapting an individual to his or her functionality, which is important for therapeutic intervention [85,86]. Other measures of resilience are directed in most cases at personality traits or adaptive processes, and not resilience in terms of an ability. Meanwhile, aspects of an ability type may prove to be particularly valuable when diagnosing people who are sick or exposed to long-term stress—in this case, assessing a specific regenerative ability may be more important than assessing resilience in terms of characteristics. Hence, in the sample of parents who have children with special educational needs, meaning that they are confronted with probably increased levels of stress, it can be assumed with a high degree of caution that their appropriate adoption in the case of lower resilience would be one of the ways to support existing resilience resources, which is important for the parents' health or psychosocial functioning. The key in this respect will be conducting research aimed at understanding the development of resilience as an adaptive ability. In this context, the immunization mechanism is the opposite of the sensitization mechanism and it describes the ability of some people to better support accumulated traumatic episodes. This suggests an improvement of the resilience ability and thus confirms that resilience has a dynamic character and can be shaped with (for

example) intervention measures [23,87]. This type of research includes limitations that constitute a challenge for future research using BRS.

These studies are not free of limitations. The first possible limitations relate to the nature of the sample in Study 2. The study group only consisted of students aged between 20 and 29. In addition, the size of the sample in this study group was significantly smaller than in study groups 1 and 3. In further studies, it is worth extending the study group to make it more representative in terms of gender and age. Another limitation is related to the results of Study 2 on the convergent and divergent accuracy of the BRS scale. Not all expected relationships have been confirmed, and that is why, in further studies, it is worth verifying the relationship between BRS and other measures of coping.

In summary, the Polish version of BRS is a reliable and accurate way of measuring resilience as the ability to bounce back from adversity and overcome various challenges or stressors. The Polish version of BRS is characterized by high reliability and accuracy when analysed based on data obtained from numerous and heterogeneous tests. This scale may be used for both research and intervention purposes.

## Supporting information

**S1 File. Polish version of the Brief Resilience Scale.**
(PDF)

## Author Contributions

**Conceptualization:** Karol Konaszewski, Małgorzata Niesiobędzka, Janusz Surzykiewicz.

**Formal analysis:** Karol Konaszewski, Małgorzata Niesiobędzka, Janusz Surzykiewicz.

**Methodology:** Karol Konaszewski, Małgorzata Niesiobędzka, Janusz Surzykiewicz.

**Validation:** Karol Konaszewski, Małgorzata Niesiobędzka, Janusz Surzykiewicz.

**Writing – original draft:** Karol Konaszewski, Małgorzata Niesiobędzka, Janusz Surzykiewicz.

**Writing – review & editing:** Karol Konaszewski, Małgorzata Niesiobędzka, Janusz Surzykiewicz.

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
