## [Decision Letter · Decision Letter 0]

9 Jul 2020

PONE-D-20-14199

Validation of the Polish version of the Brief Resilience Scale (BRS)

PLOS ONE

Dear Dr. Konaszewski,

Thank you for submitting your manuscript to PLOS ONE. After careful consideration, we feel that it has merit but does not fully meet PLOS ONE’s publication criteria as it currently stands. Therefore, we invite you to submit a revised version of the manuscript that addresses the points raised during the review process.

We look forward to receiving your revised manuscript.

Kind regards,

Frantisek Sudzina

Academic Editor

PLOS ONE

Journal Requirements:

Reviewers' comments:

Reviewer's Responses to Questions

**Comments to the Author**

1. Is the manuscript technically sound, and do the data support the conclusions?

Reviewer #1: Yes

Reviewer #2: Yes

2. Has the statistical analysis been performed appropriately and rigorously? 

Reviewer #1: Yes

Reviewer #2: Yes

3. Have the authors made all data underlying the findings in their manuscript fully available?

Reviewer #1: Yes

Reviewer #2: Yes

4. Is the manuscript presented in an intelligible fashion and written in standard English?

Reviewer #1: Yes

Reviewer #2: Yes

5. Review Comments to the Author

Reviewer #1: This is an interesting article that introduced new scientific value and deserves to be published in PLOS One.

The article consists of three independent studies on large cohorts. It deals with the subject of resilience and focuses on the Polish adaptation of the scale to measure this construct. I recommend accepting the manuscript after minor corrections:

- The authors should systematize the name of the variable. Currently, two terms are used in the work: resilience and resiliency. Consistent terminology would allow for better understanding of the text.

- My suggestions is that the "Data Analysis" section should not indicate hypotheses ("we expected..."). This section should focus primarily on the description of data analysis methods. Please consider describing the study objectives and hypotheses in a different place of the text – e.g. in the first paragraph describing the study.

- The authors should translate foreign literature items into English.

Reviewer #2: The authors have produced an excellent manuscript. The chosen topic contributes significantly to the scientific field of psychology, positive psychology, psychometry and psychology. Furthermore, it is important to note that it makes an important contribution, presenting a scale to measure resilience in a specific population.

Next, all those aspects that give quality to the work are indicated:

- Appropriate title. Descriptive and simple. Suitable for a manuscript.

- Very complete and well written introduction. It includes the most relevant aspects of the theoretical field of resilience. The writing is good.

- The object of the study is well defined.

- The studios are designed correctly. The results show the usefulness of said adaptation.

- The analysis carried out is adequate for the research question and the stated objectives.

- The discussion is elaborated correctly, although it is one of the aspects that requires more attention.

- The References are written in the correct format. Although it is recommended to include more current references.

Specifically, some of the aspects that deserve attention have to do with:

Introduction: On the one hand, authors should include more citations to support the different definitions they propose. Evidence that has no scientific support is made in the first paragraph of the introduction. The same happens in the third paragraph section that refers to "Researching abilities" the authors should complete with current references that accompany and give value to the statements they present. On the other hand, the authors should refer to the theory of coping strategies, being these relevant to understand the definition of resilience presented here. The authors could quote:

- Lazarus, R. S., y Folkman, S. (1984). Stress, Appraisal, and Coping. New York, NY: Springer.

- Cantero-García, M., & Alonso-Tapia, J. (2018). Coping and Resilience in Families With Children With Behavioral Problems. Revista de Psicodidáctica (English ed.), 23(2), 153-159.

- Alonso-Tapia, J., Rodríguez-Rey, R., Garrido-Hernansaiz, H., Ruiz, M., y Nieto, C. (2016). Coping assessment from the perspective of the person-situation interaction development and validation of the Situated Coping Questionnaire for Adults (SCQA). Psicothema, 28(4), 479-486. http://dx.doi.org/10.7334/psicothema2016:19

- Villasana, M., Alonso-Tapia, J., y Ruiz, M. (2016). A model for assessing coping and its relation to resilience in adolescence from the perspective of “person–situation interaction”. Personality and Individual Differences, 98, 250–256.

- http://dx.doi.org/10.1016/j.paid.2016.04.053

Methodology: Authors should delve further into sociodemographic characteristics. As well as clarifying why one works with such a wide age range. Furthermore, it would be advisable to expand the n of the study 2. In addition, it is necessary to clarify how the data collection was carried out. The fit indexes of the model are very good, although it is not clear why the choice is made to separate the positive and negative items. Better justify the structural model. It is unclear whether the item scores were treated as categorical-ordered or continuous variables. The choice of estimation procedure should be justified.

Discussion. You point out the methodological and theoretical limitations in a clearer way.

6. PLOS authors have the option to publish the peer review history of their article (what does this mean?). If published, this will include your full peer review and any attached files.

Reviewer #1: **Yes: **Sebastian Skalski

Reviewer #2: **Yes: **María Cantero García

---

## [Author Response · Author response to Decision Letter 0]

15 Jul 2020

Dear Frantisek Sudzina

Academic Editor

Thank you for your letters and the opportunity to revise our paper on ‘Validation of the Polish version of the Brief Resilience Scale (BRS)’. The suggestions offered by the reviewers have been immensely helpful, and we also appreciate your insightful comments on revising the paper. 

I have included the reviewers comments immediately after this letter and responded to them individually, indicating exactly how we addressed each concern or problem and describing the changes we have made. The revisions have been approved by all authors and I have again been chosen as the corresponding author. The changes are marked in yellow in the paper as you requested, and the revised manuscript is attached. We also corrected the manuscript according to Journal Requirements.

We thank you for your continued interest in our research.

Sincerely,

Answers:

Reviewer #1:

Specifically, some of the aspects that deserve attention have to do with:

1. Introduction: On the one hand, authors should include more citations to support the different definitions they propose. Evidence that has no scientific support is made in the first paragraph of the introduction. The same happens in the third paragraph section that refers to "Researching abilities" the authors should complete with current references that accompany and give value to the statements they present. On the other hand, the authors should refer to the theory of coping strategies, being these relevant to understand the definition of resilience presented here. The authors could quote:

- Lazarus, R. S., y Folkman, S. (1984). Stress, Appraisal, and Coping. New York, NY: Springer.

- Cantero-García, M., & Alonso-Tapia, J. (2018). Coping and Resilience in Families With Children With Behavioral Problems. Revista de Psicodidáctica (English ed.), 23(2), 153-159.

- Alonso-Tapia, J., Rodríguez-Rey, R., Garrido-Hernansaiz, H., Ruiz, M., y Nieto, C. (2016). Coping assessment from the perspective of the person-situation interaction development and validation of the Situated Coping Questionnaire for Adults (SCQA). Psicothema, 28(4), 479-486. http://dx.doi.org/10.7334/psicothema2016:19

- Villasana, M., Alonso-Tapia, J., y Ruiz, M. (2016). A model for assessing coping and its relation to resilience in adolescence from the perspective of “person–situation interaction”. Personality and Individual Differences, 98, 250–256.

- http://dx.doi.org/10.1016/j.paid.2016.04.053

Answer 1: Thank you for suggestion devoted the scientific suport made in the introduction. We thoroughly read the suggested literature and completed references. 

2. Methodology: Authors should delve further into sociodemographic characteristics. 

Answer 2: We added sociodemographic characteristics in the study 3.

3. As well as clarifying why one works with such a wide age range. Furthermore, it would be advisable to expand the n of the study 2.

Answer 3: A wide age range was in the study 1 and 3, except for study 2. The study group only consisted of students aged between 20 and 29 and it was a convenient group. We agree that the size of the sample in study 2 should be extended what we suggested in the paragraph devoted limitations. 

4. In addition, it is necessary to clarify how the data collection was carried out. 

Answer 4: We added characteristic of CAWI method to clarify the data collection method in study 3.

5. The fit indexes of the model are very good, although it is not clear why the choice is made to separate the positive and negative items. Better justify the structural model. 

Answer 5: Thank you for suggestion, we added more information about the tested model in the study and tried to explain more comprehensively the motives underyling the choice of the structural model.

6. It is unclear whether the item scores were treated as categorical-ordered or continuous variables. 

Answer 6:The item scores were treated as categorical-ordered variables. 

7. The choice of estimation procedure should be justified.

Answer 7:Thank you for suggestion. We added information about ML procedure.

8. Discussion. You point out the methodological and theoretical limitations in a clearer way.

Answer 8: We added section „limitations”.

Reviewer #2:

1. The authors should systematize the name of the variable. Currently, two terms are used in the work: resilience and resiliency. Consistent terminology would allow for better understanding of the text. 

Answer 1: We agree, it has been corrected.

2. My suggestions is that the "Data Analysis" section should not indicate hypotheses ("we expected..."). This section should focus primarily on the description of data analysis methods. Please consider describing the study objectives and hypotheses in a different place of the text – e.g. in the first paragraph describing the study.

Answer2: Thank you for suggestion, but we decided not to change the structure of the text. 

3. The authors should translate foreign literature items into English.

Answer 3: Thank you for suggestion - it has been corrected.

---

## [Editor Report · Decision Letter 1]

20 Jul 2020

Validation of the Polish version of the Brief Resilience Scale (BRS)

PONE-D-20-14199R1

Dear Dr. Konaszewski,

We’re pleased to inform you that your manuscript has been judged scientifically suitable for publication and will be formally accepted for publication once it meets all outstanding technical requirements.

Kind regards,

Frantisek Sudzina

Academic Editor

PLOS ONE
---

## [Editor Report · Acceptance letter]

24 Jul 2020

PONE-D-20-14199R1 

Validation of the Polish version of the Brief Resilience Scale (BRS) 

Dear Dr. Konaszewski:

I'm pleased to inform you that your manuscript has been deemed suitable for publication in PLOS ONE. Congratulations! Your manuscript is now with our production department. 

Kind regards, 

on behalf of

Dr. Frantisek Sudzina 

Academic Editor

PLOS ONE